# Field-resilient supercurrent diode in a multiferroic Josephson junction

Hung-Yu Yang [1] ✉, Joseph J. Cuozzo[2,3], Anand Johnson Bokka [1,4], Gang Qiu [1], Christopher Eckberg [1], Yanfeng Lyu[5], Shuyuan Huyan [6], Ching-Wu Chu [6,7], Kenji Watanabe [8], Takashi Taniguchi [9] & Kang L. Wang [1] ✉

The research on supercurrent diodes has surged rapidly due to their potential applications in electronic circuits at cryogenic temperatures. To unlock this functionality, it is essential to find supercurrent diodes that can work consistently at zero magnetic field and under ubiquitous stray fields generated in electronic circuits. However, a supercurrent diode with robust field tolerance is currently lacking. Here, we demonstrate a field-resilient supercurrent diode by incorporating a 2D multiferroic material into a Josephson junction, and observed a pronounced supercurrent diode effect at zero magnetic field. More importantly, the supercurrent rectification persists over a wide and bipolar magnetic field range beyond industrial standards for field tolerance. By theoretically modeling a multiferroic Josephson junction, we unveil that the interplay between spin-orbit coupling and multiferroicity underlies the unusual field resilience of the observed diode effect. This work introduces multiferroic Josephson junctions as a new field-resilient superconducting device for cryogenic electronics.

Semiconductor diodes are fundamental electronic components crucial for rectifying, regulating, and controlling the flow of electrical current in electronic circuits and systems, playing a pivotal role in the functionality of a wide range of devices from power supplies to digital electronics[1]. Supercurrent diodes, which rectify the zero-resistance supercurrent in superconductors, play key functions in digital electronics at cryogenic temperatures. For example, in an electronic flip-flop memory, a binary bit can be represented by the current going through one arm or the other; this can be achieved similarly by placing supercurrent diodes on each arm and controlling their rectification directions[2]. More importantly, for a cryogenic memory application, the readout can be done through the supercurrent diode effect (SDE) that,

in principle, leads to low power consumption and an infinite on/off ratio, thanks to the zero resistance in the superconducting state[2-4].

In the past few years, supercurrent diodes have been found extensively in various systems under a magnetic field[5-13] while only a few work at zero magnetic field. Among the zero-field supercurrent diodes[14-21], most of them require a magnetic field to polarize the ferromagnetic component and initialize the diode; the ferromagnetism grants the field-tunability to these diodes, while also makes them unable to work persistently over bipolar magnetic fields. For practical applications, ubiquitous stray fields in a common circuit environment (up to 10 mT) can easily flip the supercurrent rectification direction and make this type of diode unreliable[22]. Currently, a clear strategy for

[1]Department of Electrical and Computer Engineering, University of California, Los Angeles, CA, USA. [2]Materials Physics Department, Sandia National Laboratories, Livermore, CA, USA. [3]Department of Physics, The University of Texas at El Paso, El Paso, TX, USA. [4]Department of Materials Science and Engineering, University of California, Los Angeles, CA, USA. [5]School of Science, Nanjing University of Posts and Telecommunications, Nanjing, China. [6]Department of Physics and Texas Center for Superconductivity, University of Houston, Houston, TX, USA. [7]Lawrence Berkeley National Laboratory, Berkeley, CA, USA. [8]Research Center for Electronic and Optical Materials, National Institute for Materials Science, Tsukuba, Japan. [9]Research Center for Materials Nanoarchitectonics, National Institute for Materials Science, Tsukuba, Japan. ✉e-mail: hungyuyang@ucla.edu; wang@ee.ucla.edu

field-resilient supercurrent diodes that can work at zero magnetic field and tolerate stray fields in electrical circuits remains lacking.

The SDE is governed by the symmetry properties; the breaking of inversion and time-reversal symmetries simultaneously is essential for SDE, regardless of the material platform[23–25]. For example, a 2D superconductor with Rashba spin-orbit coupling (RSOC) breaking the inversion symmetry, and an applied in-plane transverse magnetic field breaking time-reversal symmetry, exhibits SDE[23]. In this study, we employed $NiI_2$, a 2D multiferroic material, in a van der Waals (vdW) Josephson junction (JJ) to create a field-resilient supercurrent diode. The coexisting spiral magnetic order and ferroelectric order in $NiI_2$ naturally break both inversion and time-reversal symmetry (Fig. 1a)[26–29], presumably satisfying symmetry requirements for SDE. Furthermore, the coupling between magnetic and electric orders makes a multiferroic more robust against the magnetic field (e.g., coercivity enhancement)[30,31], granting the field-resilience for SDE. Lastly, the strong magnetoelectric coupling in multiferroics enables controllable switching of magnetic order[32–34] and potentially the switching of SDE by electrical gates. Incorporating this non-volatility and gate tunability into supercurrent diodes could open the door to practical cryogenic memory devices.

## Results

### Zero-field SDE in a multiferroic vdW JJ

Since the multiferroic order in $NiI_2$ persists down to the 2D monolayer (ML) limit[35–37], we exfoliated a $NiI_2$ flake of 4 MLs thick to facilitate the Josephson coupling while keeping the multiferroic order. It is then reassembled with two $NbSe_2$ flakes to make a $NbSe_2/NiI_2/NbSe_2$ vertical vdW JJ ($NiI_2$ JJ in short), thanks to the freedom to manipulate vdW

materials with the 2D transfer assembly technique (Fig. 1b, see also Methods and Supplementary Fig. 1). Figure 1c shows a typical $V − I$ characteristic of the $NiI_2$ JJ. The quantities relevant to SDE are the critical currents for opposite bias directions, $I_{c+}$ and $I_{c−}$, at which the JJ transitions from a superconducting state to a normal state. The critical current difference $\Delta I_c \equiv I_{c+} − |I_{c−}| = −118\,\mu A$ and the diode rectification efficiency $\eta \equiv \frac{I_{c+} − |I_{c−}|}{I_{c+} + |I_{c−}|} \sim −8\%$ were obtained. The difference in magnitude also allows us to define a diode working range (gray stripes in Fig. 1c), within which the supercurrent only flows in one direction but not the other (Fig. 1d). The consistent switching with repetitive current biasing cycles (see also Supplementary Fig. 2) shows the robustness of the SDE in the $NiI_2$ JJ at zero field.

To further confirm the zero-field SDE in the multiferroic $NiI_2$ JJ, a $NbSe_2$/few-layer graphene/$NbSe_2$ vdW JJ (Gr JJ) was fabricated as a reference device, and different tests were performed to confirm the observed zero-field SDE is intrinsic (Fig. 1e). If the heating effect is significant, the rectification efficiency should flip sign between these two measurements[18]. We found that the diode rectification efficiency $\eta$ of the Gr JJ stayed near zero and that of $NiI_2$ JJ remained ~−8%, showing the heating effect was insignificant in the SDE of both devices.

Next, opposite in-plane fields $(\pm H_\parallel)$ were applied to train the magnet and devices, and then $V − I$ characteristics were measured at zero field. Here, we utilized the following fact to simulate the effect of stray fields: With a large positive (negative) magnetic field up to several tesla being applied through the magnet of our measurement system, a small negative (positive) remnant field on the order of ~1 mT could remain after the field is set to zero[38]. For a field-resilient supercurrent diode, $\eta$ must not flip its sign for opposite training fields to continue rectifying supercurrent in the same direction under stray fields. As

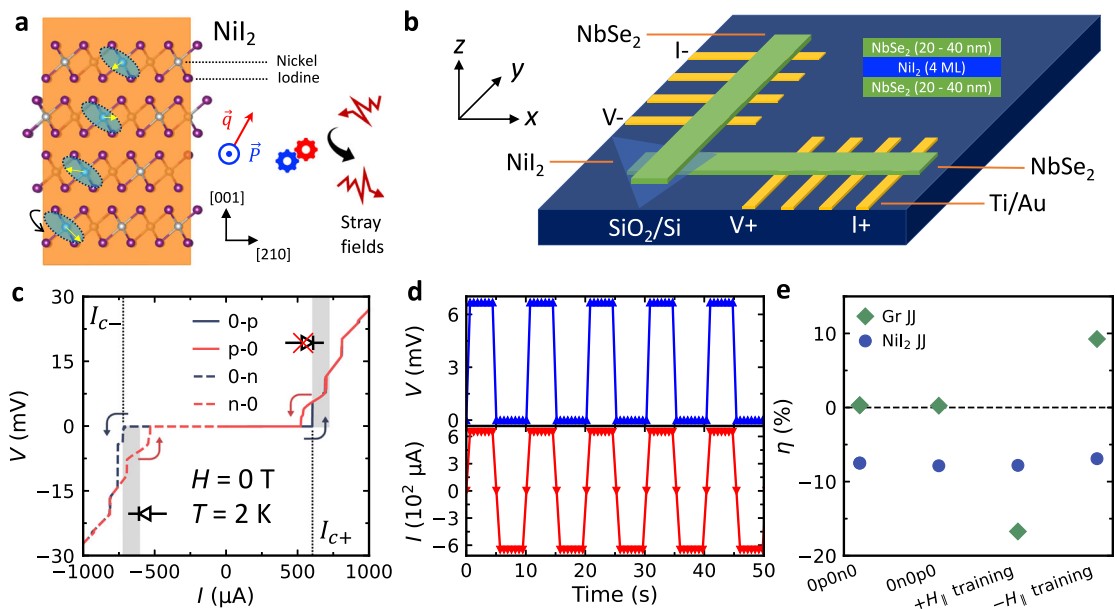

**Fig. 1 | 2D Multiferroic $NiI_2$ and zero-field supercurrent diode effect in the $NiI_2$ van der Waals (vdW) Josephson junction (JJ). a** Crystal structure and multiferroic order of $NiI_2$, which consists of a spiral magnetic order (described by the wave vector $\vec{q}$[28]) and an in-plane ferroelectric order ($\vec{P}$[29]). The yellow arrow on the Ni atoms represents the spin direction, and the shaded area represents the spin spiral plane. **b** Device geometry of the $NiI_2$ JJ with a 4 monolayer (ML) $NiI_2$. **c** $V − I$ characteristic of the $NiI_2$ JJ. 0-p, p-0, 0-n, and n-0 refer to curves with current sweeping from $0\,\mu A$ to $+1000\,\mu A$, $+1000\,\mu A$ to $0\,\mu A$, $0\,\mu A$ to $−1000\,\mu A$, and $−1000\,\mu A$ to $0\,\mu A$. The critical currents $I_{c+}$ ($600\,\mu A$) and $I_{c−}$ ($718\,\mu A$) are defined by the first critical jump in $V$ in the 0-p and 0-n (switching) curves, respectively. The gray-shaded area denotes the diode working range defined by $I_{c+}$ and $I_{c−}$.

**d** Demonstration of supercurrent rectification with $I_{bias} = \pm 650\,\mu A$. **e** Comparison of zero-field supercurrent diode rectification efficiency ($\eta = \frac{I_{c+} − |I_{c−}|}{I_{c+} + |I_{c−}|}$) between the $NbSe_2$/few-layer graphene/$NbSe_2$ JJ (Gr JJ) and the $NiI_2$ JJ under different current-sweeping and field-training protocols. The magnetic field was set to oscillate to zero from 3 T before performing the 0p0n0 and 0n0p0 measurements. The measurements for 0p0n0 and 0n0p0 tests are repeated five times to acquire error bars, which are smaller than the marker size for both cases. The 0p0n0 and 0n0p0 refer to opposite current-sweeping protocols, where a positive bias current is applied first in the 0p0n0 measurement and a negative bias current is applied first in the 0n0p0 measurement, respectively. The training fields $\pm H_\parallel = \pm 1$ T were used for both devices. The field training was performed at $T = 10$ K.

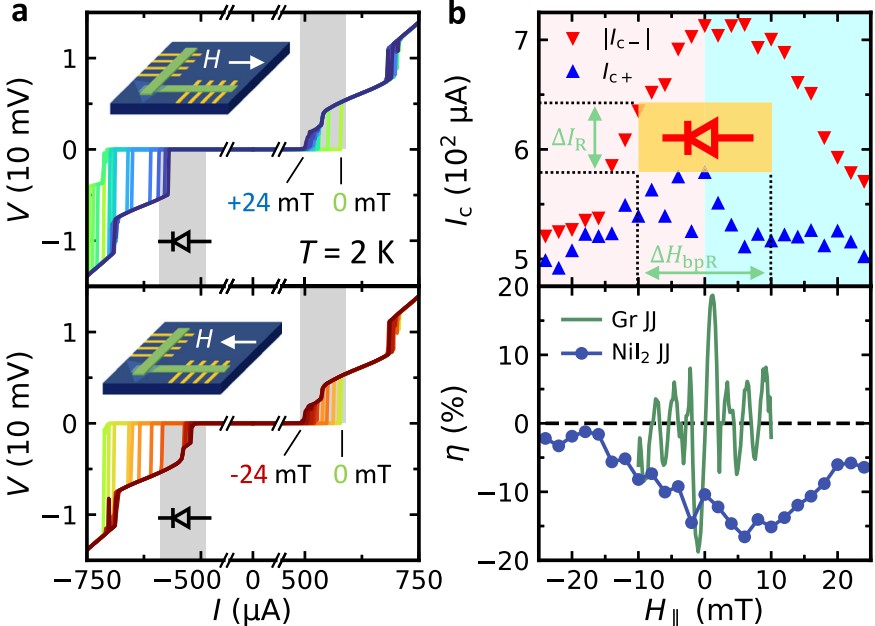

**Fig. 2 | In-plane field dependence of the supercurrent diode effect in the NiI$_2$ JJ.**
**a** Top panel: $V - I$ characteristic of the NiI$_2$ JJ with 0 mT $< H_\parallel <$ 24 mT, with a 2 mT field increment. Bottom panel: $V - I$ characteristic of the NiI$_2$ JJ with -24 mT $< H_\parallel <$ 0 mT. All plotted curves are switching curves (0-p and 0-n sweeps). The gray-shaded area shows the range over which $I_{c+}$ at different fields is distributed. The same range, while placed on the negative $I$ side for both positive and negative $H_\parallel$, mostly falls in the superconducting range, suggesting a field-symmetric supercurrent diode effect. **b** Top panel: critical current $I_{c+}$ and $|I_{c-}|$ as a function of $H_\parallel$. The pink and cyan background represents the negative and positive field range, respectively. The yellow block marks the bipolar working field range of the supercurrent diode between $\pm 10$ mT with a figure of merit $F_R = \Delta I_R \times \Delta H_{bpR} \sim 10^3$ mT $\cdot \mu A$ (see the main text for their definitions). Bottom panel: $\eta$ as a function of $H_\parallel$ of Gr JJ and NiI$_2$ JJ. The fluctuations at low fields of the NiI$_2$ JJ are likely due to the flux creep/escape effect of our magnet[38].

shown in Fig. 1e, the $\eta$ of NiI$_2$ JJ surprisingly remained negative $\sim -8\%$ after both $\pm H_\parallel$ training, in strong contrast to the Gr JJ, where its $\eta$ flipped the sign between the $\pm H_\parallel$ training. As will be discussed in Fig. 2, our Gr JJ exhibits a pronounced SDE with anti-symmetric field dependence, with $\eta \sim \pm 20\%$ for $H_\parallel \sim \pm 1$ mT, respectively. The nonzero but opposite $\eta$ values of the Gr JJ after $\pm H_\parallel$ training are thus false-positive zero-field SDE and are a result of remnant fields induced by the field training, in agreement with Fig. 2. The tests and comparison demonstrated in Fig. 1e show that the zero-field SDE in the multiferroic NiI$_2$ JJ is not only intrinsic but also field-resilient. In the NiI$_2$ JJ, $\eta$ also surpasses the values reported among the systems that do not require a field initialization for the zero-field SDE ($\eta_{max} \sim 3\%$ in both Fe(Te,Se)/Fe(Te,Se) vdW JJ[14] and NbSe$_2$/Nb$_3$Br$_8$/NbSe$_2$ vdW JJ[18]).

**Field resilience of the SDE in NiI$_2$ JJ**
To demonstrate the robustness of zero-field SDE in the NiI$_2$ JJ under stray fields, we measured the field dependence of the SDE. The $V - I$ characteristics of the NiI$_2$ JJ were acquired with in-plane fields ($H_\parallel$) ranging from $+24$ mT to 0 mT (Fig. 2a, top panel), and 0 mT to $-24$ mT (bottom panel), over which the multiferroic order persists (Supplementary Fig. 3). A consistent negative diode rectification efficiency was observed over $H_\parallel = \pm 24$ mT, with negative bias-induced critical transitions occurring beyond the gray stripe (Fig. 2a), defined as the range of critical transitions for positive bias. The unidirectional supercurrent rectification is directly linked to an unusual field dependence of SDE, which has a minute antisymmetric component and a dominant symmetric component. Such a field dependence defies the typical anti-symmetric field dependence of SDE, where an external magnetic field solely controls the time-reversal symmetry.[5,6,10,21]

The dominant symmetric in-plane field dependence is further shown by extracting $I_{c+}$ and $|I_{c-}|$ at each field (Fig. 2b, top panel) and calculating their corresponding $\eta$ (bottom panel). Again, the data points representative of $|I_{c-}|$ are always above $I_{c+}$ between $H_\parallel = \pm 24$ mT,

showing a robust negative SDE regardless of reversing the magnetic field direction. The symmetry $I_{c+} (+H_\parallel) = |I_{c-}| (-H_\parallel)$ is expected for supercurrent diodes supported by magnetochiral anisotropy, but our NiI$_2$ JJ shows a distinct field dependence owing to a different mechanism responsible for non-reciprocity. Importantly, $\eta$ consists of a predominantly symmetric, dome-shaped field dependence. Such an unusual field dependence makes it possible to draw the widest bipolar diode working range reported so far over $\pm 10$ mT ($\sim 8000$ A/m, the maximum field tolerance of industrial MRAM devices manufactured by Everspin[39]), where we can use the same amount of current biased in the opposite directions to rectify the supercurrent. Thus, a bipolar figure of merit can be defined as $F_R \equiv \Delta I_R$(current rectification range) $\times \Delta H_{bpR}$(bipolar field rectification range), as the area of the yellow block shown in Fig. 2b. For our device, $F_R$ over $\pm 10$ mT is on the order of $10^3$ mT $\cdot \mu A$, which is two orders of magnitude larger than the existing supercurrent diode where a bipolar diode working range may be defined (the maximum of $F_R$ in NbSe$_2$/Nb$_3$Br$_8$/NbSe$_2$ JJ is about $10^1$ mT $\cdot \mu A$[18]).

We highlight the unique field-resilient SDE in the NiI$_2$ JJ by comparing its field dependence to the Gr JJ (Fig. 2b, bottom panel). For the Gr JJ, the SDE therein exhibits an anti-symmetric field dependence of $\eta$ with multiple sign changes, corresponding to the lobes of the Fraunhofer interference pattern (see Supplementary Fig. 4a). The maxima of $I_c$ for opposite current biases shift from zero to opposite fields due to the self-field effect induced by the cross-junction geometry, which further leads to an SDE[40,41]. We expect the field-anti-symmetric SDE *without* a bipolar working range to be typical of vdW JJ with a non-magnetic barrier and a cross-junction geometry. On the contrary, the interference pattern of the NiI$_2$ JJ was "truncated" for the positive current bias, while preserved for the negative current bias (see Supplementary Fig. 4b). The pattern thus leads to a persistent negative $\eta$, with a dominant symmetric field dependence and a wide bipolar diode working range establishing the field-resilient SDE in NiI$_2$. The general

anti-symmetric component is also present in $NiI_2$ JJ due to the similar cross-junction geometry, but it only leads to a slight tilt towards the positive field of the dome-shaped field dependence, which remains predominantly symmetric. We have also examined the SDE under out-of-plane magnetic fields and observed again a persistent negative $\eta$ in $NiI_2$ JJ with reduced efficiency, contrary to the reference device showing a sign change of $\eta$ as the field direction is flipped (Supplementary Fig. 5). Despite having a weaker SDE under an out-of-plane magnetic field, the field-resilient nature of the SDE in the $NiI_2$ JJ remains and is distinguished from the reference device.

### Non-monotonic temperature dependence of SDE

Finally, we investigate the temperature dependence of SDE in the $NiI_2$ JJ. The results reveal its non-monotonic temperature dependence and a sign change. The zero-field SDE at different temperatures from the $V - I$ characteristics is illustrated in Fig. 3a. In Fig. 3b, the 0-p and 0-n sweeps are compared to show their critical transitions for $T \leq 5$ K. The critical transition defining $I_c$ at each temperature is labeled by short black lines, where the same transition can be tracked up to $T = 5$ K, the transition temperature of the JJ (see also Supplementary Fig. 6).

Figure 3c presents the temperature dependence of $I_{c\pm}$ of the $NiI_2$ JJ at zero field, from which $\eta$ and $|\eta_{max}|$ are extracted and compared to that of the Gr JJ measured at $H_\parallel = -1$ mT in Fig. 3d. For the Gr JJ, the SDE follows a monotonic temperature dependence where $\eta_{max}$ happens at the lowest temperature reached, similar to other non-multiferroic lateral JJs[9,21]. However, in the $NiI_2$ JJ, two unusual qualitative behaviors appear. First of all, $\eta_{max}$ appears at $T = 2.5$ K, instead of $T = 2$ K, which is the lowest temperature reached. Secondly, after the enhancement of SDE at $T = 2.5$ K, $\eta$ drops more quickly than expected and undergoes a sign change before it completely vanishes. Below, we develop a theoretical model to capture our findings of SDE in the $NiI_2$ JJ, including its appearance at zero field, enhanced bipolar field resilience, and uncommon non-monotonic temperature dependence.

### Theoretical modeling

In our $NiI_2$ JJ, electrons in $NbSe_2$ can experience spin-orbit interactions that are intrinsic to $NbSe_2$ and $NiI_2$ or arise from interfacial effects[36,42]. The geometry of the cross junction modifies the supercurrent density to reside near the surfaces of the two crossed $NbSe_2$ flakes (Fig. 4a)[43], which will enhance the role of RSOC in the Josephson coupling between the $NbSe_2$ flakes. For generality, we focus on the role of RSOC in a generic cross JJ with a helimagnet weak link[44–46]. For a propagation vector $\mathbf{q} = (q_x, q_y, 0)$, with helimagnet spin texture in real space given by $\mathbf{M} = M(-\sin(\mathbf{q} \cdot \mathbf{r}), \cos(\mathbf{q} \cdot \mathbf{r}), 0)$, we can write the Bogoliubov de-Gennes Hamiltonian in momentum space as

$$H_{BdG} = \frac{1}{2} \sum_{\mathbf{k}} \psi_{\mathbf{k}}^\dagger \begin{pmatrix} h(\mathbf{k}) - \mu & \Delta_{sc} \\ \Delta_{sc}^* & \mu - T^{-1}h(\mathbf{k})T \end{pmatrix} \psi_{\mathbf{k}} \qquad (1)$$

$$h(\mathbf{k}) = \frac{\hbar^2(k^2 + \mathbf{q}^2/4)}{2m^*} + \frac{\hbar^2(\mathbf{q} \cdot \mathbf{k})}{2m^*}\sigma_z + J_{exc}\sigma_y + \alpha_R(k_y\sigma_x - k_x\sigma_y), \qquad (2)$$

where $\psi_{\mathbf{k}} = (c_{\mathbf{k}\uparrow}, c_{\mathbf{k}\downarrow}, -c_{-\mathbf{k}\downarrow}^\dagger, c_{-\mathbf{k}\uparrow}^\dagger)^T$ is a spinor of electron creation (annihilation) operators $c_{\mathbf{k}\sigma}^\dagger$ ($c_{\mathbf{k}\sigma}$) with momentum $\mathbf{k}$ and spin $\sigma$, $\Delta_{sc} = \Delta e^{i\phi}\sigma_x$ is the superconducting gap with phase $\phi$, $\mu$ is the chemical potential, $\hbar$ is Planck's constant divided by $2\pi$, $m^*$ is the effective electron mass, $\alpha_R$ is the RSOC strength, and $J_{exc}$ is the exchange interaction energy. Here, $\sigma_i$ are Pauli matrices and $T = i\sigma_y K$ is the time-reversal operator with the complex conjugation operator $K$. In Eq. (2), the exchange spin splitting $J_{exc}\sigma_y$ arises from $M > 0$ breaking time-reversal symmetry (TRS) and the $(\mathbf{q} \cdot \mathbf{k})$ term is associated with the spin-orbit coupling induced by the spin texture $\mathbf{M}(\mathbf{r})$. The spin polarization of the exchange spin splitting term is determined by the form of $\mathbf{M}(\mathbf{r})$. Here we use an $\mathbf{M}(\mathbf{r})$ consistent with the spin texture in

$NiI_2$[35–37]. We discretize the Hamiltonian in Eq. (1) and perform numerical simulations of a helimagnetic JJ shown in Fig. 4b. Using tight-binding simulations, we calculate the Andreev bound state spectrum of the JJ to find its current-phase relationship (CPR). To model the $NiI_2$ tunnel barrier, we include a potential barrier $h_b = U_{barrier}\, \delta(x)$. Additional details are described in the Methods section.

In Fig. 4c, we present the CPR with and without RSOC for $\mathbf{q}$ oriented along $\mathbf{x}$ and $\mathbf{y}$. Here we assume $T = 0$ unless otherwise stated. The global extrema of the CPR correspond to $I_{c\pm}$. We see $I_{c+} = |I_{c-}|$ when $\alpha_R = 0$. The absence of SDE when $\alpha_R = 0$ is due to the exchange interaction generating an effective Zeeman field anti-commuting with the spin-orbit interaction originating from the spiral spin texture (second term in Eq. (2)). Since the broken TRS and inversion symmetries correspond to terms in the Hamiltonian having orthogonal spin polarization, a non-reciprocal supercurrent cannot develop[46]. When $\alpha_R > 0$, this no longer holds, and an SDE develops with a maximum efficiency of ~6% when $\mathbf{q}\|\mathbf{y}$. Thus, the combination of helimagnetism and RSOC in the JJ is sufficient to result in a zero-field SDE. We have also performed a phenomenological depairing momentum analysis that leads to a similar conclusion (see Methods and Supplementary Fig. 10 for details).

Next, we consider the effects of an external magnetic field and discuss how a symmetric-in-field SDE generally emerges in a helimagnetic JJ. To simulate the effects of a magnetic field, we consider an additional term in Eq. (2): $h_Z = g\mu_B(\mathbf{B} \cdot \sigma) = \Delta_Z/2(\hat{\mathbf{B}} \cdot \sigma)$, where $\hat{\mathbf{B}}$ is a unit vector parallel to $\mathbf{B}$. When the helimagnet spin texture and RSOC coexist in the JJ, $\eta$ is an *even* function of $\Delta_Z$ if $h_Z$ anti-commutes with the terms in the Hamiltonian that are: (i) linear in $\mathbf{k}$ parallel to the current direction and (ii) proportional to $J_{exc}$. When $h_Z$ obeys both anti-commutation relations, the BdG spectral gap closes symmetrically with $\pm \Delta_Z$ and $\eta$ is a purely even function of $h_Z$ (see "Methods"). Indeed, as calculated in Fig. 4d, a Zeeman splitting along the current direction ($\mathbf{B} = B\mathbf{x}$) leads to a *symmetric* modulation of $\eta$ in $\Delta_Z$, i.e., a symmetric-in-field SDE, regardless of the orientation of $\mathbf{q}$. For $\mathbf{B} = (0, B_y, B_z)$, $\eta$ will generally have a mixed functional dependence on the applied field (Supplementary Fig. 7). However, we emphasize that the symmetric field dependence is ubiquitous in helimagnetic JJs and is the key to the field resilience in $NiI_2$ JJ (see also Supplementary Fig. 10). This is in strong contrast to other sources of non-reciprocal switching currents associated with magnetochiral anisotropy (MCA)[7,8,12], finite-momentum superconductivity[23,25,47], Meissner currents[10,24] or self-field effects[41,43,48], which are predicted to result in antisymmetric field dependence of $\eta$ as we demonstrated in the Gr JJ reference device.

The simulated temperature scaling of $\Delta I_c$ at zero field for helimagnetic JJs is presented in Fig. 4e; the simulations reveal a non-monotonic behavior regardless of the direction of $\mathbf{q}$. We note that in our simulations we ignore thermal effects associated with domain rearrangement in $NiI_2$ and other structural changes since we are concerned with temperatures well below the Curie temperature of $NiI_2$. We also note that the temperature scale we use in simulations corresponds to temperatures $T \lesssim 1$ K, which are below the experimental conditions. Due to computational limitations on simulating the actual device size and geometry, we analyze possible sources of non-monotonic temperature scaling in our minimal model and set aside a more detailed quantitative model for future study. The exchange interaction associated with the helimagnet pushes the JJ close to a 0-$\pi$ transition, where a significant second harmonic contribution develops (orange bands in Fig. 4f) due to Andreev bound state energy level crossings at zero energy[49]. Since these states lie near zero energy, their contribution to the CPR is more quickly washed out at finite temperatures compared to even lower energy states (blue bands in Fig. 4f), which favor a $\phi = 0$ ground state. The competition between the supercurrent carried by states near zero energy and that by lower states leads to the non-monotonic scaling shown in Fig. 4e. Thus, a plausible explanation for the non-monotonic temperature dependence of SDE observed in the

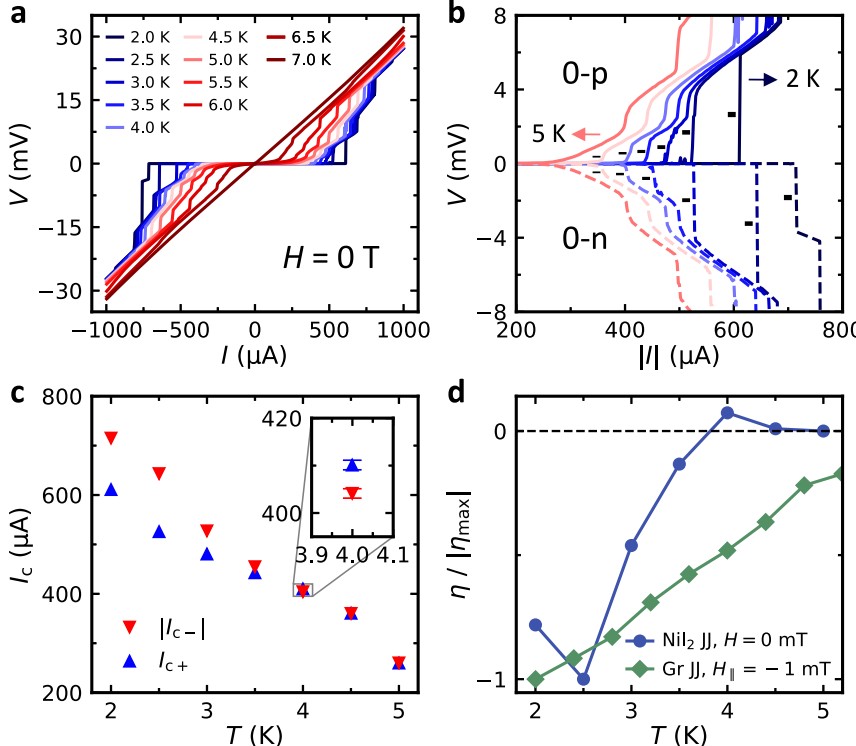

**Fig. 3 | Non-monotonic temperature dependence of supercurrent diode effect in the NiI$_2$ JJ. a** $V - I$ characteristic (switching curves) measured at different temperatures. **b** $V - |I|$ characteristic recorded at $T \leq 5$ K. The solid and dashed lines represent 0-p sweep and 0-n sweeps, respectively. The critical transitions are pointed out by short black line segments with varied widths. **c** $I_{c+}$ and $|I_{c-}|$ as a

function of temperature. The inset zooms in on the data at $T = 4$ K, where the sign change of $\eta$ appears. The error bars correspond to the 2 $\mu$A spacing between the discrete current values at which data were collected. **d** $\eta$ normalized by the maximum $|\eta_{max}|$ as a function of temperature for both NiI$_2$ JJ at zero field and Gr JJ at nonzero field. $\eta_{max}$ is $-10\%$ and $-20\%$ for NiI$_2$ and Gr JJ, respectively.

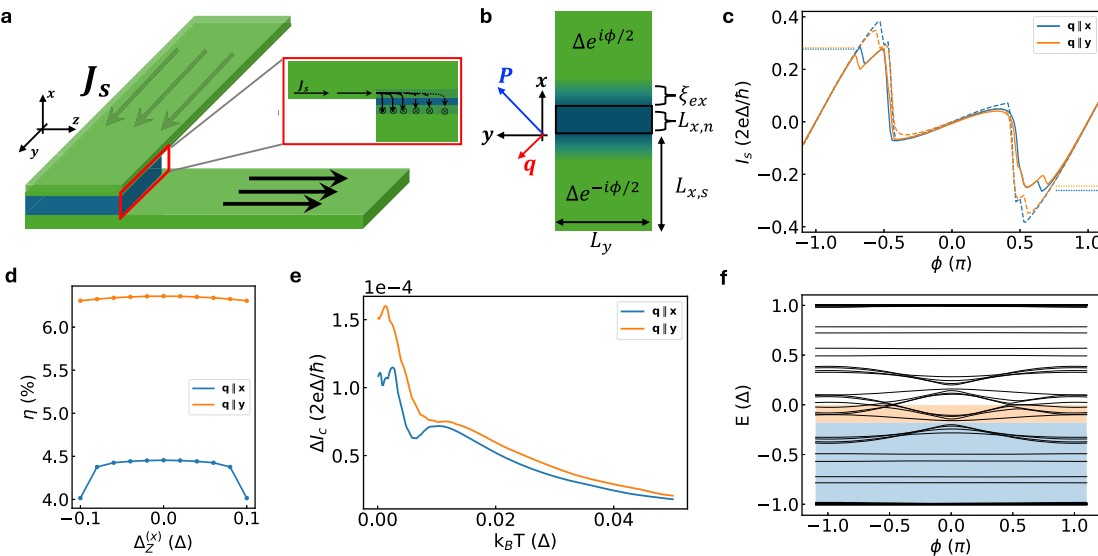

**Fig. 4 | Multiferroic JJ simulation. a** Schematic of the cross junction device where the supercurrent density $\mathbf{J}_s$ tends to reside near the surfaces of the superconducting electrodes. **b** Schematic of the planar junction corresponding to the SC/heli-magnet/SC cross-section marked by the red rectangle in panel (**a**). **c** The simulated junction CPR with (solid) and without (dashed) RSOC for $\mathbf{q}\|\mathbf{x}$ and $\mathbf{q}\|\mathbf{y}$. Unless otherwise stated, parameters used in simulations are: $\Delta = 0.4t$, $\mu = 1.57t$,

$\alpha_R = 0.004ta$, $J_{exc} = 0.3t$, $U_{barrier} = 4t$, $|\mathbf{q}| = 0.01\frac{\pi}{a}$, $L_{x,s} = 300a$, $L_{x,n} = 3a$, $L_y = 10a$, and $\xi_{exc} = 5a$, where $t = \frac{\hbar^2}{2m^*a^2}$ and $a$ is the tight-binding lattice constant. **d** Diode rectification efficiency $\eta$ versus Zeeman splitting along $\mathbf{x}$ with RSOC for $\mathbf{q}\|\mathbf{x}$ and $\mathbf{q}\|\mathbf{y}$. **e** Critical current difference $\Delta I_c = I_{c+} - |I_{c-}|$ versus temperature with RSOC. **f** The simulated Andreev bound state spectrum for $\mathbf{q}\|\mathbf{y}$ and $B = 0$.

experiment is thermal fluctuations preferentially washing out supercurrent carried by Andreev bound states responsible for higher harmonics of the CPR because of exchange interactions in $NiI_2$. It is noted that the scaling behavior depends on the details of Andreev bound states and may be modified as other parameters change (see Supplementary Fig. 8)[50,51].

Lastly, we discuss the effects of an electric polarization **P** in a multiferroic JJ from our simulations (see "Methods" and Supplementary Fig. 9 for details). Between $\pm \mathbf{P} \| y$ we see a change in the CPR, indicating that the SDE can be tuned by flipping **P**. The latter case occurs because flipping **P** simultaneously flips **q** along the current direction, whose effect on the SDE is also revealed in Fig. 4d. In addition, it is noted that the ferroelectric order in $NiI_2$ is closely linked to the strong SOC of iodine atoms, which could enhance the SDE in multiferroic JJs. Our simulation suggests that tuning electric polarization could uniquely manipulate and enhance SDE in multiferroic JJs.

Our work presents the first demonstration of a field-resilient supercurrent diode meeting an industrial standard of field-tolerance ($\pm 10$ mT) using a multiferroic $NiI_2$ vdW JJ. The key observation lies in the supercurrent diode that operates persistently not only at zero field but also under bipolar magnetic fields, matching with industrial standards for field tolerance. This invention overcomes the significant limitation in conventional supercurrent diodes, which are driven by external magnetic fields and are susceptible to disruption by stray fields. Our simulations qualitatively capture the main observations of zero-field SDE, field-resilient SDE, and non-monotonic temperature dependence of the SDE in $NiI_2$ JJ. Our theoretical modeling suggests that the combination of RSOC with helimagnetism plays a key role in the SDE in $NiI_2$ JJ, and these features may prevail in helimagnetic JJs. We point out the possibility of manipulating and enhancing the SDE by electrical gating in multiferroic JJs, which is an exciting tuning knob to explore in the future. The discovery may lead to the technology development of multiferroic supercurrent diodes with high field tolerance and tunability, and can be combined with strategies for enhancing diode efficiency to open up new possibilities for practical applications in cryogenic electronic circuits.

## Methods

### Crystal growth
Single crystals of $NiI_2$ were grown by the chemical vapor transport technique. The starting materials were mixed in a stoichiometric ratio (Ni: $I_2 = 1$: 1, 500 mg in total) and sealed in 7-inch long silica tubes under vacuum. The tubes were placed in a single-zone tube furnace, with one end at the center. The temperature of the furnace was set to 580 °C at 3 °C/minute, dwelt for 60 h, and then set to room temperature at the same rate. Black single crystals formed at the cold end of the tubes as hexagonal thin flakes. X-ray diffraction patterns (Bruker D8 ECO) of the single crystals showed a clear (003) characteristic peak at $2\theta = 13.42$ degrees, in agreement with the crystal structure reported in the Inorganic Crystal Structure Database (ICSD).

High-quality $NbSe_2$ single crystals were prepared using the iodine vapor transport method[52]. Stoichiometric amounts of Nb (99.9%; Alfa Aesar) and Se (99.5%; Alfa Aesar) powders were sealed in an evacuated quartz tube (1/2 inch diameter) with 2 mg/cm³ of iodine as the transport agent, and introduced horizontally into a tube furnace. The temperature was slowly increased to 725 °C, maintained for 3 days, and followed by furnace cooling down to room temperature. The large platelet single crystal picked out from the resulting sample was sealed on one side in another quartz tube along with a new mixture of Nb, Se, and iodine on the other side and heated through the same procedure. Subsequently, large and high-quality $NbSe_2$ single crystals were obtained.

### Device fabrication
The bottom contact electrodes were fabricated on $SiO_2$/Si substrates using photolithography and electron beam evaporation (Ti/Au, 10/40 nm). Thin flakes of h-BN, $NbSe_2$, and $NiI_2$ were exfoliated onto $SiO_2$/Si substrates using Scotch tape. To assemble the $NiI_2$ vdW JJ, a piece of h-BN was first picked by the dry transfer technique[14,53], using polypropylene carbonate (PPC) polymer spin-coated on polydimethylsiloxane (PDMS) as the stamp. Once the h-BN is picked up, other flakes of the JJ were picked up in the following order: top $NbSe_2$, $NiI_2$, and then bottom $NbSe_2$. Occasionally, the structure that the h-BN has picked up may be released on the next target flake and be heated to increase the cohesion between different layers to facilitate the pick-up process. After the entire stack of flakes was completed, the h-BN/$NbSe_2$/$NiI_2$/$NbSe_2$ vdW JJ structure was released on the bottom contacts and was ready for transport measurements without further fabrication. During the dry transfer process, the JJ area was covered by h-BN the entire time to prevent contamination due to polymer residue. The graphite/$NiI_2$/graphite tunnel junction was fabricated in the same way. For the Gr JJ and $NbSe_2$/$NbSe_2$ devices, the bottom $NbSe_2$ flakes were exfoliated onto $SiO_2$/Si substrates using Scotch tape. Other flakes were exfoliated on PDMS and then transferred on top of the bottom $NbSe_2$ flake one by one. After the entire device stack was completed, it was picked up using PPC and released on the bottom contacts. The transfer process was performed in an Ar-filled glove box with $H_2O$ and $O_2$ levels below 1 ppm using a home-built transfer stage. The two-point contact resistance between different pins was below 100 ohms.

### Transport measurements
Resistance and $V - I$ characteristics were measured in a physical property measurement system (PPMS, Quantum Design Inc.). The temperature dependence of resistance was taken with the low-frequency lock-in technique (< 10 Hz) with a 2 $\mu$A AC current excitation. For the $V - I$ characteristics, DC voltages were measured by a Keithley 2182 nanovoltmeter and a DC current bias was applied by a Keithley 6221 current source. The critical current was extracted by first taking the derivative of $V$ v.s. $I$ data, and the current value corresponds to the first peak in $dV/dI$ v.s. $I$ was marked as the critical current. If the transitions were sharp, a constant cutoff voltage may be applied to extract the critical currents. We found that the choice of extraction methods does not lead to a significant difference in the interpretation of the data. The switching curves were employed to extract critical current, unless stated otherwise. Before zero-field measurements, the magnetic field was set to 1 T and then oscillated to zero above the transition temperature of $NbSe_2$ to minimize the effect of the remnant field on the device behavior.

### Numerical simulations
We simulate the current-phase relationship (CPR) of the multiferroic JJ using the following tight-binding Bogoliubov-de Gennes Hamiltonian:

$$H^{(BdG)} = \sum_{\mathbf{r}_n} \psi_{\mathbf{r}_n}^\dagger \left[ \left( (4 + q^2/4)t - \mu + U_{dip}(\mathbf{r}_n) + U_{barrier}\delta_{x_n,0} \right) \tau_z \otimes \sigma_0 \right] \psi_{\mathbf{r}_n}$$
$$+ \sum_{\mathbf{r}_n} \psi_{\mathbf{r}_n}^\dagger \left[ \frac{\Delta_Z^{(x)}(\mathbf{r}_n)}{2} \tau_z \otimes \sigma_x + J_{exc}(\mathbf{r}_n)\tau_0 \otimes \sigma_y - \Delta(\mathbf{r}_n)\tau_y \otimes \sigma_y \right] \psi_{\mathbf{r}_n}$$
$$+ \sum_{<\mathbf{r}_n, \mathbf{r}_m>} \delta_{y_n, y_m} \psi_{\mathbf{r}_n}^\dagger \left( -t\tau_z \otimes \sigma_0 + i\alpha_R \tau_0 \otimes \sigma_x + itq_y h(\mathbf{r}_n, \mathbf{r}_m)\tau_0 \otimes \sigma_z \right) \psi_{\mathbf{r}_m}$$
$$+ \sum_{<\mathbf{r}_n, \mathbf{r}_m>} \delta_{x_n, x_m} \psi_{\mathbf{r}_n}^\dagger \left( -t\tau_z \otimes \sigma_0 - i\alpha_R \tau_z \otimes \sigma_y + itq_x h(\mathbf{r}_n, \mathbf{r}_m)\tau_0 \otimes \sigma_z \right) \psi_{\mathbf{r}_m},$$

$$(3)$$

where $\psi_{\mathbf{r}_n} = (c_{\mathbf{r}_n\uparrow}, c_{\mathbf{r}_n\downarrow}, c_{\mathbf{r}_n\uparrow}^\dagger, c_{\mathbf{r}_n\downarrow}^\dagger)^T$, and $c_{\mathbf{r}_n,\rho\sigma}^\dagger$ ($c_{\mathbf{r}_n,\rho\sigma}$) is the creation (annihilation) operator for an electron at site $\mathbf{r}_n$ with spin $\sigma$, and $\tau_i$ and $\sigma_i$ are Pauli matrices. The JJ is defined by the regions $-L_{x,s} - L_{x,n}/$

$2 \leq x \leq L_{x,s} + L_{x,n}/2$ and $-L_y/2 \leq y \leq L_y/2$. Then we have

$$U_{dip}(\mathbf{r}) = \begin{cases} \kappa_{dip}(\mathbf{P} \cdot \mathbf{r}), & -L_{x,n}/2 \leq x \leq L_{x,n}/2 \\ 0, & otherwise \end{cases}$$

$$\Delta_Z^{(x)}(\mathbf{r}) = \begin{cases} \Delta_Z^{(x)}, & -L_{x,n}/2 \leq x \leq L_{x,n}/2 \\ 0, & otherwise \end{cases}$$

$$J_{exc}(\mathbf{r}) = \begin{cases} J_{exc} & |x| \leq L_{x,n}/2 + \xi_{exc} \\ 0, & otherwise \end{cases} \qquad (4)$$

$$\Delta(\mathbf{r}) = \begin{cases} \Delta e^{i\phi/2}, & x < -L_{x,n}/2 \\ \Delta e^{-i\phi/2}, & x > L_{x,n}/2 \\ 0, & otherwise \end{cases}$$

$$h(\mathbf{r}_n, \mathbf{r}_m) = \begin{cases} 1 & |x_n|, |x_m| \leq L_{x,n}/2 + \xi_{exc} \\ 0, & otherwise \end{cases},$$

where $\xi_{exc}$ is the characteristic length scale of the exchange proximity effect in the superconducting leads. To calculate the CPR, we diagonalize the tight-binding Hamiltonian to solve for eigenvalues $\{\epsilon_n(\phi)\}$. At temperature $T$, the CPR for a short ballistic junction is ref. 54

$$I_s(\phi, T) = -\frac{2e}{\hbar} \sum_n \tanh\left(\frac{\epsilon_n}{2k_B T}\right) \frac{d\epsilon_n}{d\phi}. \qquad (5)$$

In our simulations, we take the superconducting gap to be constant and only consider $k_B T \leq 0.05\Delta$, where the suppression of $\Delta$ according to BCS theory is negligible. Using a lattice constant of 10 nm and an effective electron mass of 0.03 times the bare electron mass, the superconducting gap is estimated to be 5 meV. This is certainly larger than the gap in NbSe$_2$, but it allows for more faster simulations without compromising the qualitative accuracy of our results. None of our conclusions about the zero field SDE, bipolar field resilience, or non-monotonic T dependence are changed by considering a smaller superconducting gap in our simulations. Simulation results are presented in Supplementary Figs. 7 and 8.

Here, we discuss the effect of an electric polarization **P** in a multiferroic JJ on its SDE. Numerically, we approximate the electric polarization with an effective dipole approximation resulting in an electric potential $U_{dip} = \kappa_{dip}(\mathbf{P} \cdot \mathbf{r})$, where $\kappa_{dip}$ characterizes the permeability of the multiferroic layer. Owing to the perfect conductivity of the superconducting electrodes, we consider an electric polarization confined to the normal region of the JJ. Furthermore, we constrain $\mathbf{P} \times \mathbf{q} \parallel + \mathbf{z}$ as is required for spin-spiral multiferroic ordering. Supplementary Fig. 9a-c show the Andreev bound state spectra of the multiferroic JJ with RSOC. Supplementary Fig. 9d shows the $T = 0$ CPR with $|\mathbf{P}| > 0$. At $T = 0$ and for $\pm \mathbf{P} \parallel x$, we find that flipping the sign of $\mathbf{P} \parallel x$ does not affect the diode rectification efficiency or polarity. Here, the asymmetry introduced in the junction by **P** leads to an asymmetric normal resistance, similar to typical ferroelectric diodes.[55] On the other hand, for $\pm \mathbf{P} \parallel y$, flipping **P** results in a change in the CPR since **q** along the current direction is simultaneously flipped. In general, the tunability of the CPR with **P** depends on the details of the junction and a more systematic study is needed to determine how to optimize the electric tunability of the CPR by manipulating **P**.

### Depairing momentum analysis

We can consider the heuristic argument given by Yuan and Fu[23] to explore the diode effect in the superconducting helimagnet as it relates to finite Cooper pair momentum associated with a current bias. Consider the effect of a depairing momentum $\ell$ on the energy spectrum of the superconducting helimagnet, where the

Hamiltonian in Eq. (2) is replaced by

$$h_{BdG}(k, \ell) = \begin{pmatrix} h(k + \ell/2) - \mu & \Delta \\ \Delta & \mu - T^{-1}h(k - \ell/2)T \end{pmatrix}. \qquad (6)$$

Here we focus on $\ell = \ell_x \mathbf{x}$. The key to the heuristic argument given by Yuan and Fu for a Rahsba superconductor with an in-plane Zeeman field is that an asymmetry in the closing of the spectral gap (manifestation of the diode effect) arises when the Zeeman field is *perpendicular* to $\ell$ (i.e., current direction). Mathematically, this condition is a consequence of the form of the spin-orbit coupling, e.g., for $\ell = \ell_x \mathbf{x}$, the spin-orbit term and Zeeman terms in the Hamiltonian are aligned $\sim \left(\ell_x + \Delta_Z^{(y)}\right)\sigma_y$, effectively shifting the depairing momentum, see Supplementary Fig. 10a, b. Hence, if the depairing momentum term in the spin-orbit interaction is perpendicular to the Zeeman field, then there is no asymmetry in the closing of the spectral gap with $\ell$. Given the general form of the spin-orbit interaction and effective Zeeman splitting in a helimagnet in the absence of an external magnetic field, we see that it is not possible to observe an asymmetry in the closing of the spectral gap with $\ell$. This implies centrosymmetric superconducting helimagnets generally will not show a diode effect associated with a depairing momentum mechanism. Now, if we consider Rashba spin-orbit coupling (RSOC) as discussed in the main text, we find that a non-reciprocal critical current develops in our simulations. Incorporating RSOC into the helical superconductivity analysis above, the SDE in the depairing momentum emerges as an asymmetric suppression of the gap with the depairing momentum, see Supplementary Fig. 10c–f.

To picture the even-in-$H$ SDE observed in the experiment, it's helpful to consider how $h_z$ affects the spectral gap of a superconducting helimagnet with RSOC. Introducing a depairing momentum into the BdG Hamiltonian will result in an indirect gap closing in the dispersion, as discussed above. Phenomenologically, the indirect gap closing in superconductors with SDE will be asymmetric in the depairing momentum[23]. Now, the Zeeman effect tends to suppress the spectral gap of a superconductor with spin-singlet pairing due to a spin population imbalance. In our case, we find SDE will be symmetric in $\Delta_Z$ when $h_z$ causes the spectral gap to close *directly*, i.e., it does not contribute to an indirect spectral gap suppression, favoring a finite depairing momentum. This is shown explicitly in Supplementary Fig. 10g–j, where the effect of the depairing momentum is symmetric in $\Delta_Z^{(x)}$ whereas this ideal symmetry is lifted with $\Delta_Z^{(z)}$.

## Data availability
The data that support the findings of this study are available within the article and its Supplementary Information files. All data generated from this study are available from the corresponding authors upon request.

## Code availability
The codes that support the findings of this study are available from the corresponding authors upon request.

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

## Acknowledgements

H.Y.Y. thanks Fazel Tafti at Boston College for generously providing access to his laboratory facilities for the growth of $NiI_2$ single crystals. H.Y.Y. thanks Margarita Davydova, Adolfo O. Fumega, Yi Tseng, Connor A. Occhialini, Jonathan Gaudet, and Ilya Sochnikov for the fruitful discussions. J.J.C. thanks Enrico Rossi for helpful discussions and François Léonard for his critical reading of the manuscript. K.L.W. acknowledges the support of the U.S. Army Research Office MURI program under Grants No. W911NF-20- 2-0166 and No. W911NF-16-1-0472. J.J.C. is supported by an LDRD. We acknowledge the use of the Nano and Pico Characterization Lab in the California NanoSystems Institute at UCLA. Sandia National Laboratories is a multi-mission laboratory managed and operated by National Technology & Engineering Solutions of Sandia, LLC (NTESS), a wholly owned subsidiary of Honeywell International Inc., for the U.S. Department of Energy's National Nuclear Security Administration (DOE/NNSA) under contract DE-NA0003525. This written work is authored by an employee of NTESS. The employee, not NTESS, owns the right, title and interest in and to the written work and is responsible for its contents. Any subjective views or opinions that might be expressed in the written work do not necessarily represent the views of the U.S. Government. The publisher acknowledges that the U.S. Government retains a non-exclusive, paid-up, irrevocable, worldwide license to publish or reproduce the published form of this written work or allow others to do so, for U.S. Government purposes. The DOE will provide public access to results of federally sponsored research in accordance with the DOE Public Access Plan. K.W. and T.T. acknowledge support from the JSPS KAKENHI (Grant Numbers 21H05233 and 23H02052) and World Premier International Research Center Initiative (WPI), MEXT, Japan. Y.L. acknowledges the support from the National Natural Science Foundation of China (grant No. 12104238). We acknowledged the use of the software VESTA[56] for drawing the atomic structure shown in Fig. 1.

## Author contributions

H.Y.Y. and K.L.W. conceived the project and designed the experiments. H.Y.Y. synthesized the $NiI_2$ crystals. Y.L., S.H., and C.W.C. synthesized the $NbSe_2$ crystals. T.T. and K.W. provided and characterized bulk h-BN crystals. H.Y.Y. and C.E. performed atomic force microscopy measurements. G.Q. fabricated the bottom contact electrodes. H.Y.Y. fabricated the $NiI_2$ JJ, $NbSe_2/NbSe_2$, and graphite/$NiI_2$/graphite devices. A.J.B. and H.Y.Y. fabricated the Gr JJ device. H.Y.Y. performed the transport measurements. H.Y.Y. and J.J.C. analyzed the transport data. J.J.C. developed the theoretical model and performed the numerical simulations. H.Y.Y., J.J.C., and K.L.W. wrote the manuscript with inputs from all authors.

## Competing interests

The authors declare no competing interests.
