## [Transparent Peer Review file · Nature Communications]

Field-Resilient Supercurrent Diode in a Multiferroic Josephson Junction

Corresponding Author: Professor Kang Wang

Version 0:

Reviewer comments:

Reviewer #1

(Remarks to the Author)

The manuscript shows a systematic series of experiments while showing a magnetic-field robust vdW Josephson junction showing superconducting diode effect. The modelling of the multiferroic helimagnet as a weak-link of a vertical Josephson junction also adds fresh perspective to future investigations of current-phase relationship of Josephson junctions with interesting weak-links. Although several external factors can give false-signatures of superconducting diode effect, the authors have carefully investigated and ruled many of them with careful protocols and test case such as Graphene reference junction.

However, I would like to have the authors opinion on the following comments/questions before I recommend the publication of this manuscript in Nature Communications:

1. In Fig. 2b, the I_{c+} (blue legend) seem to fluctuate a lot in the range of +10 mT to -10 mT. Author should provide an explanation for this behaviour. Also, what accounts for the pronounced difference in shape between I_{c+} and I_{c-} ? A link from the proposed theory will be helpful.
2. In Fig. 3, the non-monotonic dependence of diode rectification efficiency on temperature has been attributed to 0- π transitions. I believe this claim must be solidified by having more points around the transition point, thus clearly indicating a smooth 0- π transition. Additionally, the errors in the measurements (maybe by error bars/confidence bands) should be added to rule out the possibility of this particular point being a deviation.
3. In extended Fig 4b, there is no change of the dV/dI value across the Cyan guiding line for the NiI2 JJ, as the resistance should be finite beyond the Fraunhofer line. The truncation of the curve at higher fields due to interlayer Josephson coupling should be elaborated in details.
4. In line 224, it has been said that the Rashba spin-orbit-coupling (RSOC) and the effective Zeeman field being perpendicular can't produce SDE, but there have been reports of current, SOC and Zeeman field being orthogonal to each other to produce SDE (Jeon et al, ref 21). A comparative discussion pertaining to this might be useful.
5. Line 146; Could the term "Symmetric" be elaborated? The dome-shaped diode-rectification efficiency (bottom panel in Fig. 2b) attains a zero value about -15mT whereas it doesn't do the same in the other side of Fig 2b.

Reviewer #2

(Remarks to the Author)

Yang et al, observed an intrinsic zero-field superconducting diode effect (zero-field SDE) in a vertical van der Waals Josephson junction NbSe₂/NiI₂/NbSe₂ made of a multiferroic NiI₂, and reported its field resilience under ubiquitous stray

fields and over bipolar magnetic fields. The experimental results are supported by theoretical modeling of the multiferroic vdW JJ. Theoretical calculations and numerical simulations qualitatively capture the role of multiferroicity with helical spin textures and its interplay with Rashba spin-orbit coupling, unveiling the origin of zero-field SDE, its field resilience, and its non-monotonic temperature dependence.

Although zero-field SDE has already been demonstrated in various superconducting structures, zero-field SDE in a multiferroic vdW JJ could be a promising device for superconducting cryogenic technologies. In addition, the superconducting structure hosting multiferroicity could provide new grounds for exploring the coexistence of superconductivity, ferroelectricity, and magnetism with unconventional spin textures. With this fundamental importance, I am happy to recommend the manuscript entitled "Field-Resilient Supercurrent Diode in a Multiferroic Josephson Junction" [NCOMMS-24-75701-T] for publication in "Nature Communications". However, before publication I would like the authors to clarify a couple of questions.

My questions and comments are listed below:

1. Origin of SDE: Symmetry breaking and the role of field/magnetization should be further clarified:

1.1 A symmetric field dependence indicates that the observed SDE is not determined by the field or magnetization. In the exchange interaction term $J_{\text{ex}} \sigma_y$, why spin polarisation is taken along the y-axis?

1.2 As shown in Figure 2(b), the efficiency of the field-trained SDE device at $H \parallel \neq 0$ (~15–20%) is greater than the efficiency of the zero-field SDE (~8%). What could be the underlying mechanism behind the field-induced enhancement of SDE efficiency?

1.3 A symmetry-based analysis for the field effects on SDE would also help to understand the antisymmetric field dependence of SDE under out-of-plane magnetic fields (Extended Data Fig. 5). That is, could SDE under out-of-plane magnetic fields be associated with the Ising and/or Dzyaloshinskii–Moriya interactions originating from broken in-plane mirror symmetries and helical spin textures?

2. A comment on field-resilience: The authors claimed "Our work presents the first demonstration of a field-resilient supercurrent diode using a multiferroic NiI₂ vdW JJ." In principle, any zero-field SDE with a symmetric field dependence should be field resilient if nonlinear non-reciprocal behaviour remains independent of magnetic field or magnetisation, possibly due to symmetry constraints. In this context, previously proposed zero-field SDEs in time-reversal-symmetric superconducting structures [Ref.18 and Ref. 13] could also be field-resilient, though, the bipolar figure of merit FR may vary from case to case. However, SDE proposed here is indeed a first field-resilient zero-field SDE in a multiferroic system with broken time-reversal symmetry, not in general.

3. Non-monotonic temperature dependence: In superconducting diodes, non-monotonic temperature dependence could depend on the field/magnetisation strength, sample fabrication, and the strength of the disorder; see Ref. 13. In the present manuscript, as noted by the authors, non-monotonic temperature dependence is linked to the $0-\pi$ transition driven by the exchange interaction associated with helimagnetism. Does that mean a shift from field-free SDE to magnetisation-determined SDE, due to domain rearrangement and helical spin structural change?

Version 1:

Reviewer comments:

Reviewer #1

(Remarks to the Author)

Most of my questions/comments have been addressed by the authors. In particular, they have (i) provided a plausible explanation for the fluctuations in the critical current in Fig. 2b, attributing them to flux creep effect, (ii) added error bars to Fig. 3 and toned down claims regarding the $0-\pi$ transition, (iii) clarified my concern about spin-orbit coupling and their orientation with respect to effective Zeeman fields, revising the manuscript text.

I am satisfied with their responses. The manuscript is now suitable for publication.

(Remarks on code availability)

Reviewer #2

(Remarks to the Author)

The authors addressed the remarks with satisfactory arguments and I appreciate the changes/clarifications made in the revised manuscript. I therefore recommend this article for publication in Nature Communications.

(Remarks on code availability)

Reviewer #3

(Remarks to the Author)

This manuscript experimentally presents a field-resilient supercurrent diode by integrating a multiferroic material, NiI₂, into a van der Waals Josephson junction.

NiI₂ possesses the spiral magnetic order and in-plane ferroelectric order, so it breaks both inversion and time-reversal symmetries, satisfying the essential conditions for the supercurrent diode effect.

The authors experimentally demonstrate that the system indeed exhibits a superconducting diode effect at zero magnetic field, and the efficiency of the superconducting diode is about 8%.

They also show that this superconducting diode can survive on magnetic fields.

This manuscript has undergone one round of peer review.

While I find some of the authors' responses to the previous reviewers satisfactory, I have reservations about others.

Below are my questions and comments:

1. From Fig.1c, the difference between I_{c+} and $|I_{c-}|$ is minimal, resulting in a very low superconducting diode efficiency (only about 8%). Given such low efficiency, does this superconducting diode have practical application value?

2. From the manuscript, it seems that all the experimental data were obtained from a single Josephson junction device. Is that the case? Have the authors tested more Josephson junction devices? How is the reproducibility?

3. In Fig.1d and Extended Data Fig.2, the authors study that the rectification effect and show the consistent switching under repetitive current biasing cycles at a frequency of about only 0.1Hz. This frequency is too low. If the frequency reaches 0.1 GHz, can the results still hold?

4. While under out-of-plane magnetic fields, what is the magnitude of the bipolar figure of merit? It seem to be very small in this case.

5. From Fig.3d, the superconducting diode efficiency is significantly modulated by temperature. What is the underlying physical mechanism?

6. What are the temperature values in Figs. 4c and 4d? Are these calculations taken at zero temperature? Experimentally, the temperature is around 2K. Given that the superconducting transition temperature is about 10K, the temperature $k_B T$ in numerical calculations should be set to 0.2Δ . Additionally, the temperature axis (abscissa) in Fig.4e only extends to about 0.05Δ . This scale is significantly lower than experimental conditions, resulting in a mismatch with actual measurements.

7. The authors set the superconducting gap $\Delta = 0.4t$. Could they provide an estimate of the superconducting gap (e.g., in units of meV)?

8. Many acronyms (e.g., RSOC, TRS, CPR) are not defined upon their first occurrence.

So I can not recommend the manuscript for publication in the present version.

(Remarks on code availability)

Version 2:

Reviewer comments:

Reviewer #3

(Remarks to the Author)

The authors have addressed all my questions.

I would like to recommend it for publication on Nature Communications now.

(Remarks on code availability)

Response to the Comments on the Manuscript “Field-Resilient Supercurrent Diode in a Multiferroic Josephson Junction”

We thank the reviewers for their careful and prompt reviews. Below, we address the reviewers’ questions and comments on a point-by-point basis. We have incorporated their suggestions into our manuscript. In the following, the reviewers’ comments appear in *italic* font, and our responses are in **blue** regular font. A list of changes is attached at the end. The changes we made are **colored red** in the revised manuscript.

Comments from the reviewer #1

The manuscript shows a systematic series of experiments while showing a magnetic-field robust vdW Josephson junction showing superconducting diode effect. The modelling of the multiferroic helimagnet as a weak-link of a vertical Josephson junction also adds fresh perspective to future investigations of current-phase relationship of Josephson junctions with interesting weak-links. Although several external factors can give false signatures of superconducting diode effect, the authors have carefully investigated and ruled many of them with careful protocols and test case such as Graphene reference junction.

However, I would like to have the authors opinion on the following comments/questions before I recommend the publication of this manuscript in Nature Communications:

We thank the first reviewer’s appreciation of the quality of our work and the new perspective it provides for the future development in Josephson junctions. Their extensive feedback on our paper is helpful for us to improve our work. We have taken into account each comment carefully, and have incorporated them into the revised manuscript. Our responses are listed point by point below.

1. In Fig. 2b, the I_{c+} (blue legend) seem to fluctuate a lot in the range of +10 mT to -10 mT. Author should provide an explanation for this behaviour. Also, what accounts for the pronounced difference in shape between I_{c+} and I_{c-} ? A link from the proposed theory will be helpful.

We thank the reviewer for pointing out this important issue. We attribute this effect to the flux creep effect of our magnet, which has a more pronounced influence on the Fraunhofer interference patterns at small fields and/or when there is a field history before the measurement. As a comparison, we examined the data of our NbSe₂/few-layer graphene/NbSe₂ Josephson junction (Gr JJ) device, where we had a chance to conduct more measurements before the device degraded. As shown in Fig. R1 above, despite the two measurements having been conducted under the same condition with the same field sequence, there are different random jumps of critical current values (pointed out by white circles and arrows), similar to the data of NiI₂ JJ shown in Fig. 2b. For the magnet in our Quantum Design PPMS system, it is known that the trapped flux in the magnet may creep and escape to the sample space, leading to the actual magnetic field being different from the nominal applied magnetic field (see Ref. 38 of the manuscript). The data shown in Fig. R1b gives a clean example of such an effect. On the positive field side, the mapping is smooth without any fluctuations, and the fourth lobe of the Fraunhofer pattern emerges near $H_{||} \sim 5$ mT. On the negative field side, however, there is a sudden jump in the critical current near $H_{||} \sim -5$ mT and the data become discontinuous. The data after the jump mimics the latter half of the third lobe ($H_{||} \sim -4$ mT); furthermore, the fourth lobe appears at $H_{||} \sim -6$ mT, which is larger than the counterpart on the positive field side. This is not consistent with the JJ interference physics but can be explained by the flux creep and escape. The flux likely escaped the magnet and leads to an additional positive magnetic field $H_{||} \sim 1$ mT, which can explain why the data starting from $H_{||} \sim -5$ mT mimics that of $H_{||} \sim -4$ mT, and also why the fourth lobe emerges later at $H_{||} \sim -6$ mT instead of $H_{||} \sim -5$ mT as its counterpart on the positive field side. Fig. R1a shows a more complicated case which fluctuates more than once (white circle), suggesting the random nature of the flux creep/escape effect. We have noticed that the best protocol to record the

data without the influence of flux creep/escape effect is to reset the magnet before the measurement (see caption of Fig. R1 and Ref. 38), start from zero field and go unidirectionally (say sweep the field up first), then reset the magnet again, and then finish the sweep of the other direction. At the time when we record the data of NiI₂ JJ, the measurement was performed unidirectionally from positive field to negative field, which means there will be a field history and we expect the flux/escape effect to cause some fluctuations in our data. We believe that such an effect does not alter our main conclusion. We have added the field sweeping direction in the caption and reminded the reader of a potential flux creep/escape effect in our data.

On the difference in shape of $I_{c\pm}(H_{\parallel})$: there is no requirement for devices with *zero field* SDE to have symmetry in $I_{c\pm}(H_{\parallel})$ curves. For systems with SDE due to magnetochiral anisotropy, a natural symmetry is expected to exist under the transformation $I \rightarrow -I$ and $H \rightarrow -H$. In our system, magnetochiral anisotropy is not responsible for the SDE, so we don't expect such a symmetry to exist (in fact, such a symmetry is incompatible with zero field SDE).

To clarify the role of symmetry in the $I_{c\pm}$ curves, we added in line 146:

“The symmetry $I_{c+}(+H_{\parallel}) = |I_{c-}(-H_{\parallel})|$ is expected for supercurrent diodes supported by magnetochiral anisotropy, but our NiI₂ JJ shows a distinct field dependence owing to a different mechanism responsible for non-reciprocity.”

2. In Fig. 3, the non-monotonic dependence of diode rectification efficiency on temperature has been attributed to 0- π transitions. I believe this claim must be solidified by having more points around the transition point, thus clearly indicating a smooth 0- π transition. Additionally, the errors in the measurements (maybe by error bars/confidence bands) should be added to rule out the possibility of this particular point being a deviation.

We thank the reviewer for the insightful comment. We have revisited our data and add error bars to Fig. 3 in the revised manuscript. The non-monotonic behavior remains valid when error bars are included. We have also revisited our device, while the device has degraded such that the supercurrent no longer tunnels through the junction and prevents us from collecting more data of critical current at different temperatures. However, we note that beyond the sign change of diode rectification efficiency as temperature changes, our device also exhibits a steep decrease of diode rectification efficiency at $T = 3$ K, which is still 2 K away from the junction transition temperature (5 K); such an anomalous behavior is in strong contrast to our graphene reference device and other supercurrent diodes, where the diode rectification efficiency shows a smooth decrease as temperature increases and slowly reaches zero at transition temperature. We agree with the referee that more data would be needed to perform more quantitative analysis, and we have toned down the statement of this part and focus on its qualitative behavior. We hope to focus on the field-resilient nature of our device and reports this novel finding in a timely manner.

3. In extended Fig 4b, there is no change of the dV/dI value across the Cyan guiding line for the NiI₂ JJ, as the resistance should be finite beyond the Fraunhofer line. The truncation of the curve at higher fields due to interlayer Josephson coupling should be elaborated in details.

We thank the reviewer for raising the question about this peculiar behavior, which is characteristic of multilayer system where several JJs coexist and can be viewed as they are in series. Such a behavior can be seen in artificially engineered multilayers (Fig. 5 in the work by I. Nevirkovets, J. Evetts, M. Blamire, Phys. Lett. A 187, 119–126 (1994)) and 2D van der Waals JJs (Fig. 3B in Ref. 17]. For example, in twisted $\text{Bi}_2\text{Sr}_2\text{CaCu}_2\text{O}_{8+x}/\text{Bi}_2\text{Sr}_2\text{CaCu}_2\text{O}_{8+x}$ van der Waals Josephson junction, it was argued that the Josephson coupling can occur between 1) each layer of top/bottom $\text{Bi}_2\text{Sr}_2\text{CaCu}_2\text{O}_{8+x}$ flake, or 2) the gap between the top and bottom $\text{Bi}_2\text{Sr}_2\text{CaCu}_2\text{O}_{8+x}$ flakes. When both JJs are present, the overall critical current mapping will be affected by the interference patterns of both JJs, showing the truncation at higher fields since the pattern of the intrinsic JJ is slowly varying.

We note that in our work, such a behavior is present in the NiI_2 JJ but not in the Gr JJ, most likely due to the difference of Josephson penetration depths ($\sim 10 \mu\text{m}$ for the NiI_2 JJ and $\sim 400 \mu\text{m}$ for the Gr JJ) compared to the junction's lateral width (about several μm). With the Josephson penetration depth much larger than the junction width, the weak link across the few-layer graphene in the Gr JJ leads to a predominant Josephson coupling; however, with the penetration depth slightly larger than the width, the Josephson coupling across NiI_2 is more likely to be comparable to the intrinsic one from top/bottom NbSe_2 flakes, leading to a case of two JJs in series and the coexistence of two JJ patterns. We have elaborated this point and added the consideration of Josephson penetration depths when discussing Fig. 4b in its caption. A new reference (I. Nevirkovets, J. Evetts, M. Blamire, Phys. Lett. A 187, 119–126 (1994)) is also added to provide more comprehensive literature of this behavior.

4. In line 224, it has been said that the Rashba spin-orbit-coupling (RSOC) and the effective Zeeman field being perpendicular can't produce SDE, but there have been reports of current, SOC and Zeeman field being orthogonal to each other to produce SDE (Jeon et al, ref 21). A comparative discussion pertaining to this might be useful.

We thank the reviewer for allowing us the opportunity to clarify our discussion. The SOC we are discussing in line 224 is not Rashba-type SOC, but rather the second term in Eq. (2) associated with the spiral spin texture of NiI_2 . Our intent was to explain why the SDE is absent when $\alpha_R = 0$: In spin space, the SOC associated with the spin spiral texture points along the z-axis while the effective Zeeman splitting points along the y-axis. This orthogonality is general (i.e. does not depend on our choice for $\mathbf{M}(\mathbf{r})$) and leads to zero SDE, in agreement with the symmetry analysis in Ref. 46 of the revised manuscript. In hindsight, we recognize how our language was confusing, particularly in comparison to Ref. 21 where orientation of SOC is the direction of broken inversion symmetry in real space.

Considering this, we have revised the statement in line 224 (line 232 in the revised manuscript):

“The absence of SDE when $\alpha_R = 0$ is due to the exchange interaction generating an effective Zeeman field anti-commuting with the spin-orbit interaction originating from the spiral spin texture (second term in Eq. (2)). Since the broken TRS and inversion symmetries correspond to terms in the Hamiltonian having orthogonal spin polarization, a non-reciprocal supercurrent cannot develop [46].”

5. Line 146; Could the term “Symmetric” be elaborated? The dome-shaped diode-rectification efficiency (bottom panel in Fig. 2b) attains a zero value about -15mT whereas it doesn't do the same in the other side of Fig 2b.

We thank the reviewer for examining our data carefully and giving feedback for us to make the presentation of our results clearer. Indeed, the field dependence of NiI₂ JJ's diode rectification efficiency is not 100% symmetric, by which we mean $\eta(+|H_{\parallel}|) = \eta(-|H_{\parallel}|)$. Rather, it is a mixture of a symmetric and an antisymmetric component, where $\eta(+|H_{\parallel}|) = -\eta(-|H_{\parallel}|)$, and that is why the diode rectification efficiencies at +15 mT and -15mT are not the same. However, we emphasize that the unique behavior of the field-resilient supercurrent diode effect in our NiI₂ JJ device lies in that its field dependence is **largely dominated by the symmetric part**, while the antisymmetric part only plays an insignificant role. Such a dominant symmetric part allows for the diode rectification efficiency remains the same sign (negative) even when the magnetic field direction is flipped, in strong contrast to the Gr JJ reference device, where the efficiency flips its sign as soon as the field direction is flipped. Since both NiI₂ JJ and Gr JJ have the same cross-junction geometry, which is known to generate self-field that gives rise to an antisymmetric supercurrent diode effect (see Ref. 41 and 43 of the manuscript), it is expected that the diode rectification efficiency of the NiI₂ JJ device also has an antisymmetric component, despite being much smaller compared to its symmetric component. We have clarified the meaning of a symmetric field dependence in the diode rectification efficiency when describing our results in Fig. 2, and explained more about the self-field effect in Extended Data Fig. 4.

Comments from the reviewer #2

Yang et al, observed an intrinsic zero-field superconducting diode effect (zero-field SDE) in a vertical van der Waals Josephson junction NbSe₂/NiI₂/NbSe₂ made of a multiferroic NiI₂, and reported its field resilience under ubiquitous stray fields and over bipolar magnetic fields. The experimental results are supported by theoretical modeling of the multiferroic vdW JJ. Theoretical calculations and numerical simulations qualitatively capture the role of multiferroicity with helical spin textures and its interplay with Rashba spin-orbit coupling, unveiling the origin of zero-field SDE, its field resilience, and its non-monotonic temperature dependence.

Although zero-field SDE has already been demonstrated in various superconducting structures, zero-field SDE in a multiferroic vdW JJ could be a promising device for superconducting cryogenic technologies. In addition, the superconducting structure hosting multiferroicity could provide new grounds for exploring the coexistence of superconductivity, ferroelectricity, and magnetism with unconventional spin textures. With this fundamental importance, I am happy to recommend the manuscript entitled “Field-Resilient Supercurrent Diode in a Multiferroic Josephson Junction” [NCOMMS-24-75701-T] for publication in “Nature Communications”. However, before publication I would like the authors to clarify a couple of questions.

We are grateful for second reviewer's acknowledgment of our work and its fundamental importance, and the recommendation of our work for publication. We also thank the reviewer for raising important questions for us to clarify and further improve our work. We have clarified each question, and our responses can be seen point by point below.

My questions and comments are listed below:

1. Origin of SDE: Symmetry breaking and the role of field/magnetization should be further clarified:

1.1 A symmetric field dependence indicates that the observed SDE is not determined by the field or magnetization. In the exchange interaction term $J_{exc} \sigma_y$, why spin polarisation is taken along the y-axis?

The exchange interaction term in Eq. (2) has spin polarization along the y-axis because of the form of the magnetization $\mathbf{M}(\mathbf{r})$. Deriving the Hamiltonian we used in k-space from a real space Kondo lattice model with an on-site exchange term $J_{exc} \mathbf{M}(\mathbf{r}) \cdot \sigma$. Performing a unitary transformation dependent on the form of $\mathbf{M}(\mathbf{r})$ ($U = \exp \left[\frac{i(\mathbf{q} \cdot \mathbf{r}) \sigma_z}{2} \right]$) allows the Hamiltonian to be represented in momentum space (e.g. also see Ref. 44 of the revised manuscript) and determines the spin polarization of the exchange interaction term in Eq. (2) on the manuscript. Our choice for $\mathbf{M}(\mathbf{r})$ is consistent with previous reports on the magnetic texture of few-layer NiI₂ (Ref. 35-37 in revised manuscript).

To clarify the origin of the spin polarization of the exchange interaction term, we added in line 222:

“The spin polarization of the exchange spin splitting term is determined by the form of $\mathbf{M}(\mathbf{r})$. Here we use an $\mathbf{M}(\mathbf{r})$ consistent with the spin texture in NiI₂ [35-37].”

1.2 As shown in Figure 2(b), the efficiency of the field-trained SDE device at $H_{||} \neq 0$ ($\sim 15-20\%$) is greater than the efficiency of the zero-field SDE ($\sim 8\%$). What could be the underlying mechanism behind the field-induced enhancement of SDE efficiency?

We thank the reviewer for reading our data thoroughly and pointing out this peculiar behavior. We think this behavior is because the diode rectification efficiency in NiI₂ JJ has both a symmetric and an antisymmetric component. The symmetric component is the dominant contribution and leads to the field-resilient supercurrent diode effect, originating from the multiferroic JJ. The antisymmetric component comes from the self-field induced from the cross-junction geometry of our device. In Fig. 2b, we may see that the dome-shaped field dependence of the diode rectification efficiency of NiI₂ JJ has its peak shifted to the positive field leading to a larger efficiency compared to the zero-field value, and the efficiency at the negative fields has a lower efficiency compared to the zero-field value, consistent with the scenario that the diode rectification efficiency is composed of a dominant symmetric component and a less significant antisymmetric component. We have clarified both symmetric and antisymmetric components in Fig. 2b and also add more explanations under Extended Data Fig. 4.

1.3 A symmetry-based analysis for the field effects on SDE would also help to understand the antisymmetric field dependence of SDE under out-of-plane magnetic fields (Extended Data Fig. 5). That is, could SDE under out-of-plane magnetic fields be associated with the Ising and/or Dzyaloshinskii–Moriya interactions originating from broken in-plane mirror symmetries and helical spin textures?

We appreciate the reviewer’s question on the role of exchange coupling in SDE. Extended Data Fig. 5d shows an asymmetric SDE under out-of-plane magnetic fields for the NbSe₂/NbSe₂ junction and a symmetric SDE for the NiI₂ junction. For the NbSe₂/NbSe₂ junction, we suspect the asymmetric SDE is not associated with Ising spin-orbit interactions intrinsic to the two flakes due to the thickness of the flakes being much larger than those reported in Ref. 8 where an asymmetric SDE was reported in a few-layer NbSe₂ flake with an out-of-plane magnetic field. However, at the NbSe₂/NbSe₂ interface, broken in-plane mirror symmetry induces a spin-orbit interaction that leads to magnetochiral anisotropy, consistent with the symmetry requirements for SDE discussed in Ref. 47.

For the NiI₂ junction, we attribute the symmetric SDE to (1) the helical spin texture in NiI₂ which is associated with the intra- and inter-layer magnetic exchange interaction in NiI₂, and (2) interfacial effects. Like the NbSe₂/NbSe₂ junction, the NiI₂ junction breaks in-plane mirror symmetries that induce spin-orbit interactions. We demonstrate that a helimagnet spin texture combined with broken in-plane mirror symmetries (described by a weak Rashba-type spin-orbit interaction) leads to a symmetric and robust SDE for in-plane magnetic fields. For out-of-plane magnetic fields, the perfect symmetry of SDE is lifted but the diode polarity is still robust in a wide bipolar field range, see Extended Data Fig. 7. The spin texture in NiI₂ can be accurately described with a classical Heisenberg spin model including Dzyaloshinskii-Moriya and Ising interactions, but due to having many contributing factors to the exact out-of-plane magnetic field dependence of SDE, we consider a minimal model to capture a field-resilient SDE in a helimagnet Josephson junction that is consistent with the broken in-plane mirror symmetry. We plan to theoretically investigate the relationship between various spin models and SDE in future work.

2. A comment on field-resilience: The authors claimed "Our work presents the first demonstration of a field-resilient supercurrent diode using a multiferroic NiI₂ vdW JJ." In principle, any zero-field SDE with a symmetric field dependence should be field resilient if nonlinear non-reciprocal behaviour remains independent of magnetic field or magnetisation, possibly due to symmetry constraints. In this context, previously proposed zero-field SDEs in time-reversal-symmetric superconducting structures [Ref.18 and Ref. 13] could also be field-resilient, though, the bipolar figure of merit FR may vary from case to case. However, SDE proposed here is indeed a first field-resilient zero-field SDE in a multiferroic system with broken time-reversal symmetry, not in general.

We thank the reviewer for this insightful comment and bring existing relevant literatures to our attention. By “field-resilient”, we mainly would like to emphasize the wide bipolar working range of magnetic fields up to ± 10 mT in NiI₂ JJ device, meeting the industrial standards of field-tolerance for the first time. In the revised manuscript, we have toned down our statement and make sure our description respects the previous achievements in the literature.

3. Non-monotonic temperature dependence: In superconducting diodes, non-monotonic temperature dependence could depend on the field/magnetisation strength, sample fabrication, and the strength of the disorder; see Ref. 13. In the present manuscript, as noted by the authors, non-monotonic temperature dependence is linked to the $0-\pi$ transition driven by the exchange interaction associated with helimagnetism. Does that mean a shift from field-free SDE to magnetisation-determined SDE, due to domain rearrangement and helical spin structural change?

Since T in the experiment is far from the Curie temperature of NiI₂ (~ 40 K in the few-layer limit), we suspect spatial variations of the multiferroic order with increasing T is unlikely to cause the observed non-monotonicity of the diode efficiency. In our work, the non-monotonic T dependence occurs when the junction is near a $0-\pi$ transition at $T = 0$, see Fig. 4f. The temperature dependence of the current-phase relationship is described in Eq. (9), where the hyperbolic tangent tends to suppress supercurrent from low-energy Andreev bound states more quickly than higher energy states as T increases. This leads to the simulated non-monotonic diode efficiency.

To clarify the role of magnetic order in the non-monotonic temperature dependence of the diode efficiency, we added in line 266:

“We note that in our simulations we ignore thermal effects associated with domain rearrangement in NiI₂ and other structural changes since we are concerned with temperatures well below the Curie temperature of NiI₂.”

List of changes

- Caption of Fig. 2: The origin of the fluctuations in our data is clarified.
- Line 146: The field dependence of I_{c+} v.s. I_{c-} is clarified.
- Line 232: The roles of SOC and Zeeman field in our multiferroic Josephson junction are clarified.
- Fig. 3c: We add an inset to zoom in the data at $T = 4$ K, showing the sign change of ΔI_c . The error bars are defined in the caption. We have also revised our descriptions of the data and focus on the qualitative behaviors.
- Description of Fig. 2b: We clarify the symmetric and anti-symmetric component of the data presented in Fig. 2 and elaborate the meaning of “symmetric” under the section “Field resilience of the SDE in NiI₂ JJ”.
- Caption of Extended Data Fig. 4: We elaborate on the self-field effect and how it contributes to the anti-symmetric component of SDE in our devices. We also provide more explanations to clarify the coexistence of two Fraunhofer patterns as a result of two JJs in series.
- Line 222: We clarify the origin of the exchange interaction term in our model.
- Line 291: We revise the statement of our achievement to align with the literature.
- Line 266: We clarify the roles of domain rearrangement and structural changes in our system.
- New references added: I. Nevirkovets, J. Evetts, M. Blamire, Phys. Lett. A 187, 119–126 (1994).

Response to the Comments on the Manuscript “Field-Resilient Supercurrent Diode in a Multiferroic Josephson Junction”

We thank the reviewers for their careful and prompt reviews. Below, we address the reviewer #3’ questions and comments on a point-by-point basis. We have incorporated their suggestions into our manuscript. In the following, the reviewers’ comments appear in *italic* font, and our responses are in blue regular font. A list of changes is attached at the end. The changes we made are colored red in the revised manuscript.

Comments from the reviewer #3

This manuscript experimentally presents a field-resilient supercurrent diode by integrating a multiferroic material, NiI2, into a van der Waals Josephson junction. NiI2 possesses the spiral magnetic order and in-plane ferroelectric order, so it breaks both inversion and time-reversal symmetries, satisfying the essential conditions for the supercurrent diode effect. The authors experimentally demonstrate that the system indeed exhibits a superconducting diode effect at zero magnetic field, and the efficiency of the superconducting diode is about 8%. They also show that this superconducting diode can survive on magnetic fields. This manuscript has undergone one round of peer review. While I find some of the authors' responses to the previous reviewers satisfactory, I have reservations about others. Below are my questions and comments:

We thank the third reviewer’s acknowledgement of our previous responses being satisfactory, and their further insights into our work. Their perspective on our paper is helpful for us to improve our work. We have taken into account these questions and comments carefully, and have addressed them in the revised manuscript. Our responses are listed point by point below.

1. From Fig.1c, the difference between I_{c+} and $|I_{c-}|$ is minimal, resulting in a very low superconducting diode efficiency (only about 8%). Given such low efficiency, does this superconducting diode have practical application value?

We appreciate the opportunity to clarify our work's significance. Our primary achievement is not record-breaking efficiency, but record-high field resilience—a critical property for any practical electronic device. Without this resilience, even a 100% efficient superconducting diode is impractical, as its rectification direction would be constantly disrupted by the stray fields ubiquitous in electronic circuits, preventing robust memory storage. Our work presents a clear strategy to achieve this robustness by incorporating a multiferroic material into a Josephson junction for the first time.

This approach can be combined with other strategies currently being pursued to enhance efficiency. Together, these efforts will eventually enable the development of superconducting diodes that are both highly efficient and field-resilient, paving the way for their use in practical applications. In this regard, this work is a significant step toward developing practical supercurrent diodes. In the revised manuscript, we have added a remark to point out that the enhancement of both diode efficiency and field tolerance are important for practical applications.

2. From the manuscript, it seems that all the experimental data were obtained from a single Josephson junction device. Is that the case? Have the authors tested more Josephson junction devices? How is the reproducibility?

We thank the reviewer for their careful review of our data. We reported all data from a single device to ensure self-consistency. We have confirmed the reproducibility of the zero-field supercurrent diode effect and its field-resilience in another device (see below). Since we are currently investigating other aspects of this system, we hope to reserve additional details for future work due to potential competitions. We are confident that the detailed comparison between the NiI₂ JJ and the reference devices already provides strong support for the field-resilient nature of multiferroic supercurrent diodes. Due to the timely importance of our novel findings and potential competitions, we hope to focus on the field-resilient nature of our device and report our findings in a timely manner.

3. In Fig.1d and Extended Data Fig.2, the authors study that the rectification effect and show the consistent switching under repetitive current biasing cycles at a frequency of about only 0.1Hz. This frequency is too low. If the frequency reaches 0.1 GHz, can the results still hold?

We appreciate the reviewer's question regarding high-frequency applications. Although this was not our primary focus, high-frequency operation is a promising direction. For example, supercurrent diodes have reached 100 kHz (with GHz estimations) [Phys. Rev. B **107**, 054506 (2023)], and superconducting devices generally can function in the GHz range [Proceedings of the IEEE, **92**(10), 1564-1584, (2004)], suggesting that a GHz demonstration is attainable. We hope to focus the current work on the novel and critical finding of our device's field-resilience.

4. While under out-of-plane magnetic fields, what is the magnitude of the bipolar figure of merit? It seem to be very small in this case.

We thank the reviewer for pointing out the difference between the in-plane and out-of-plane field behaviors. The out-of-plane field is indeed more detrimental to the supercurrent in these junctions,

which in turn weakens the diode effect (as shown in Extended Data Fig. 5c). We did not calculate a precise figure of merit for this orientation since the diode effect quickly diminishes away from zero magnetic field and our estimation would have large errors. Our results clearly demonstrate robust in-plane field resilience and show that this property persists even for out-of-plane fields for the first time, suggesting the need to consider all field directions in future studies. To avoid confusions, we have clarified in the revised manuscript that the SDE under an out-of-plane magnetic field is weaker, but the field resilience remains.

5. From Fig.3d, the superconducting diode efficiency is significantly modulated by temperature. What is the underlying physical mechanism?

We thank the reviewer for this important question. We believe a plausible underlying physical mechanism is thermal fluctuations washing out the supercurrent carried by Andreev bound states responsible for higher harmonics in the current-phase relationship.

In our calculation presented in Fig. 4, we find the exchange interaction spin splitting and the spin-orbit coupling in Eq. (2) drives the Josephson junction near a $0-\pi$ transition. As we highlight in Fig. 4f, the Andreev bound states contribution to the CPR is dominated by states in two energy windows: $-0.2\Delta \lesssim E \leq 0$ (orange) and $E \lesssim -0.2\Delta$ (blue). The states in the orange window contribute to a π -periodic contribution to the CPR, and states in the blue window contribute a conventional 2π -periodic term. Thermal fluctuations preferentially wash out supercurrent from states near the Fermi energy, which leads to a suppression of second harmonic (π -periodic) terms in the CPR more quickly than 2π -periodic terms as T increases. Given the importance of higher harmonics in the CPR to realize a SDE, this can cause significant modulation of the diode effect by temperature seen in Fig. 4e.

To clarify the underlying physical mechanism, we added to the manuscript at line 284:

“Thus, a plausible explanation for the non-monotonic temperature dependence of SDE observed in the experiment is thermal fluctuations preferentially washing out supercurrent carried by Andreev bound states responsible for higher harmonics of the CPR because of exchange interactions in NiI_2 .”

6. What are the temperature values in Figs. 4c and 4d? Are these calculations taken at zero temperature?

We thank the reviewer for these important clarifying questions. Yes, the temperature is taken to be zero. We have updated the manuscript to convey this in line 233:

“Here we assume $T = 0$ unless otherwise stated.”

Experimentally, the temperature is around 2K. Given that the superconducting transition temperature is about 10K, the temperature $k_B T$ in numerical calculations should be set to 0.2Δ .

We agree with the reviewer that finite temperatures should be included in calculations for quantitative accuracy, but computational limitations on the system size we simulate constrain the quantitative

accuracy of our model of $\Delta I_c(T)$. For instance, our calculations include far fewer Andreev bound states than we expect to have in the actual device. Due to the relatively small number of sub-gap bound states in Fig. 4f and their energies lying primarily below 0.5Δ , most of the supercurrent in our calculations (including the interesting features associated with helimagnetism) are washed out when $k_B T = 0.2\Delta$. Our approach in the theoretical analysis was to develop a minimal model for a multiferroic Josephson junction which captures the key features observed in the experiment: zero field SDE, bipolar field resilience and non-monotonic T dependence. Calculations with this minimal model come at some cost to the quantitative accuracy.

To clarify the temperature scale in simulations, we added in line 271:

“We note that the temperature scale we use in simulations corresponds to temperatures $T \lesssim 1$ K which are below the experimental conditions.”

Additionally, the temperature axis (abscissa) in Fig.4e only extends to about 0.05Δ . This scale is significantly lower than experimental conditions, resulting in a mismatch with actual measurements.

We thank the reviewer for this comment. Indeed, the temperature scale in the theoretical simulations is smaller than the range of temperatures considered in the experiment. As mentioned above, due to the relatively small system size we simulated, the SDE is suppressed at temperatures near $0.2\Delta/k_B$, so we focused on smaller temperature scales where a non-monotonic temperature dependence of η is observed. Apart from the approximations we discussed above, the temperature scale is limited since we do not account for a suppression of the superconducting gap with T (for $k_B T < 0.05\Delta$, the superconducting gap is negligibly suppressed by T). Accounting for this suppression would require individual calculations of the bound state spectrum for each T , which is time consuming with marginal qualitative insight. Besides, as we discuss in the Theoretical Modeling and Methods sections, the temperature dependence of SDE in our simulations are sensitive to other details of the model (e.g. junction width).

Incorporation of thermal effects into our model is intended to offer a plausible explanation for the unusual non-monotonic temperature dependence observed in the experiment (discussed in response to point 5). We plan to investigate more quantitatively accurate models for multiferroic Josephson devices in the future.

To clarify the limitations of the modeling of thermal effects, we added in line 273:

“Due to computational limitations on simulating the actual device size and geometry, we analyze possible sources of non-monotonic temperature scaling in our minimal model and set aside a more detailed quantitative model for future study.”

7. The authors set the superconducting gap $\Delta = 0.4t$. Could they provide an estimate of the superconducting gap (e.g., in units of meV)?

We thank the reviewer for the opportunity to clarify parameters used in the numerical simulations. Using a lattice constant of 10 nm and an effective electron mass of 0.03 times the bare electron mass, the superconducting gap is estimated to be 5 meV. This is certainly larger than the gap in NbSe₂, but it allows for faster simulations without compromising the qualitative accuracy of our results. None of our conclusions about the zero field SDE, bipolar field resilience, or non-monotonic T dependence are changed by considering a smaller superconducting gap in our simulations.

To clarify this, we added a sentence to line 680 in the Methods section:

“Using a lattice constant of 10 nm and an effective electron mass of 0.03 times the bare electron mass, the superconducting gap is estimated to be 5 meV. This is certainly larger than the gap in NbSe₂, but it allows for more faster simulations without compromising the qualitative accuracy of our results. None of our conclusions about the zero field SDE, bipolar field resilience, or non-monotonic T dependence are changed by considering a smaller superconducting gap in our simulations.”

8. Many acronyms (e.g., RSOC, TRS, CPR) are not defined upon their first occurrence.

We thank the reviewer for carefully reading our manuscript and help us improve it by introducing acronyms. We have checked the acronyms, and the readers can now find RSOC at the line 69, CPR at line 230, and we have also introduced TRS at line 223.

List of changes

- Line 316: We emphasize the importance of both field tolerance and diode efficiency for practical applications of supercurrent diodes.
- Line 181: We clarify our finding under an out-of-plane magnetic field.
- Line 223: We introduce the acronym TRS for time-reversal symmetry.
- Line 284: We clarify the possible mechanism for non-monotonic temperature scaling of the diode effect.
- Line 233: We clarify the assumption of zero temperature in calculations presented in Fig. 4c-d.
- Line 271: We clarify the difference between the temperature scale in the simulations and experiment.
- Line 680: We clarify the parameters used in simulations.